# Intention to leave the current health facility among healthcare workers in Ethiopia: Systematic review and meta-analysis

Gizew Dessie Asres[1]*, Yeshiwork Kebede Gessesse[2], Molalign Tarekegn Minwagaw[1]

1 Amhara Public Health Institute, Bahir Dar, Amhara, Ethiopia, 2 Durbete Primary Hospital, Durbete, Amhara, Ethiopia

* gizew.dessie@gmail.com

**Data Availability Statement:** All relevant data are within the paper and its Supporting information files.

## Abstract

### Background

Strengthening workforce management to address retention challenges is worldwide concern. Ethiopia did different human resource reforms to improve retention and performance of available health workforce to step up towards universal health coverage. However, there is limited and fragmented research on intention to leave their current facility and related factors on health workers. This review was done to assess pooled national prevalence of intention to leave their current health facility and related factors among Ethiopian health workers.

### Methods

This systematic review and meta-analysis followed PRISMA guidelines. Authors prepared a review protocol per Joanna Briggs Institute (JBI) manual for evidence synthesis and got registered from PROSPERO for transparency. The authors conducted a comprehensive search of PubMed, Web of Science, Cochrane, Google Scholar and African Index Medicus databases and grey literature like WHO library from 8 June 2022 to 30 May 2023. Pooled prevalence of intention to leave current health facility and related factors was computed using MedCalk, Meta Essentials and R software. Publication bias was assessed using Egger's test and a funnel plot.

### Results

Pooled prevalence of intention to leave among health workers in Ethiopian healthcare setting was found to be 63.52% (95% CI (58.606–67.904)) for random effects model at Q = 141.5689 ($I^2$(inconsistency) = 90.82%, P < 0.0001). Only organizational justice OR = 0.29 (0.14–0.61) was found to be a significantly associated factor for health workers' intention to leave their current healthcare setting in Ethiopia.

### Conclusion

More than 6 in 10 of the health workers in Ethiopia were ready to leave their current healthcare facility. This result was higher than studies done in other parts of the world, even in

**Funding:** The authors received no specific funding for this work.

**Competing interests:** The authors have declared that no competing interests exist.

African countries. The associated factor for health workers' intention to leave their current health facility was only organizational justice. So, health authorities should improve their organizational justice to retain their employees.

## Introduction

The shortage of health workforce in healthcare settings presents a significant global challenge, exacerbated by the COVID-19 pandemic since 2019 [1]. One of the major factors contributing to this shortage at the facility level is the turnover of experienced health workers. In this context, turnover intention, also referred to as intention to leave, is crucial. While the actual turnover of the health workforce can be retrospectively calculated from annual exit data, a prospective and more actionable indicator is measuring turnover intention. Turnover intention refers to employees' willingness to leave their organization within a defined period [2–4].

The method for calculating the intention to leave and the proportion of employees with the intention to leave an organization is based on established practices in survey research and human resource management. To calculate the proportion of intention to leave, divide the number of employees who have intention to leave their organization using a survey tool by the total employees assed and multiply it by 100 to get a percentage [5, 6]. Intention to leave one's job is not exactly the same as intention to leave one's organization [7].

Turnover appears to be as simple as leaving staff members at their working facility or having them move to another healthcare facility, then filling the void in their positions with new hires who are supposed to be equally qualified. In the case of the health workforce, there is still a supply shortage of some disciplines to replace the vacant post and this creates a shortage at one health facility and may create excess at other health facilities. High turnover rates at healthcare facilities disrupt workflow [8], lower the quality of care [9], and result in significant financial costs [10]. Retaining experienced health workers in health facilities is a challenge worldwide [11].

An organization's sustainability relies heavily on its resources, particularly its human resources [12], which provide a competitive advantage [13]. Turnover, especially the voluntary departure of experienced personnel, significantly impacts service quality in healthcare facilities [14]. Intention to leave one's current facility serves as a proxy indicator for actual turnover [15, 16]. Researchers have identified several domains influencing health workforce turnover, including leadership, education, staffing levels, professional issues, support at work, personal factors, demographics, and financial remuneration [17].

Studies have reported annual turnover rates ranging from 15% to 27% for nurses in various countries [18–21]. In China, a systematic review revealed high intention to leave among primary health workers (30.4%) and general practitioners (47%), with factors including demographics, job characteristics, and job satisfaction [22, 23]. Similarly, studies in Japan and Saudi Arabia showed significant intention to leave among psychiatry nurses (41%) and pharmacists (61.9%), respectively [24, 25].

In sub-Saharan Africa, intention to leave among nurses was prevalent (18.8%–50.74%), with varying rates in different countries [26, 27]. To address these challenges, low- and middle-income countries need to strengthen institutional environments for health workforce production, deployment, retention, and performance management [11].

In Ethiopia, while initiatives have been proposed and approved by the Ministry of Health (EMoH) to enhance workforce motivation and retention, implementation remains

inconsistent across regional states, and health workers perceive the packages as inadequate and not inclusive of all cadres [28].

Despite efforts to address the issue, Ethiopia lacks comprehensive data on health workers' intentions to leave their current facilities. Previous studies have shown inconsistent prevalence rates, ranging from 30.4% to 80.6% [29, 30]. Therefore, this systematic review and meta-analysis aim to estimate the pooled prevalence of health professionals' intention to leave their current health facilities in Ethiopia and identify associated factors, utilizing studies conducted from 2013 to 2022.

### Review question

This systematic review was guided by the following review questions based on the condition, context and population (CoCoPop) framework;

1. What is the pooled prevalence of intention to leave their current health facility among health workers in Ethiopia?

2. What are the factors for the intention to leave their current health facility among health workers in Ethiopia?

3. Is there a variation of intention to leave among health workers categories in Ethiopia?

## Methods

### Search strategy and selection criteria

This systematic review and meta-analysis was conducted and reported following the Preferred Reporting Items for Systematic Reviews and Meta-Analyses (PRISMA) protocol [31]. A review protocol was developed as per the Joanna Briggs Institute (JBI) manual for evidence synthesis to pre-define the objectives and methods of the systematic review [32]. The protocol details the criteria the reviewers have used to include and exclude studies. This protocol was registered at the International Prospective Register of Systematic Reviews (PROSPERO) ID = CRD42022312559.

We conducted a comprehensive search of PubMed, Web of Science, Cochrane, Google Scholar and African Index Medicus databases and grey literature like the WHO library from 8 June 2022 to 30 May 2023. The search terms used at PubMed were 'Health worker' OR 'health staff' OR 'healthcare staff' OR 'human resource for health' OR 'health employee' OR 'health professional' AND 'health care setting' OR health facility OR health institution OR hospital OR health centre OR clinic OR health post AND determinants OR factors OR predictors AND intention to leave OR turnover intention OR intention to quit AND Ethiopia. In addition, reference listing techniques from bibliographic search results and grey literatures were used.

### Eligibility criteria

The inclusion criteria were based on CoCoPop mnemonic (Condition, Context, and Population) and used for reviewers to screen and select articles for prevalence data on intention to leave their current health facility and related factors (Table 1).

### Study screening and selection

The process of screening was conducted in two levels. Level one was based on the titles and abstracts of the paper. In the second level, papers that have passed the screening level were

**Table 1. Inclusion and exclusion criteria of studies, 2023.**

| Included studies | Excluded studies |
|---|---|
| 1. Studies were done on health workers' intention to leave their current facility between 2013 and 2022 in Ethiopia | 1. Health workers of the Woreda Health Office, Zonal Health Department, Regional Health Bureau, Ministry of Health, blood bank, Ethiopian Food Drug Administration, professional associations and health science colleges |
| 2. Observational studies with analytical parts. | 2. Intention to leave studies done on health workers in health care settings outside of Ethiopia |
| 3. Intention to leave Studies done at all levels of health care settings or facilities. | 3. Intention to leave studies done on health workers in health care settings before 2013 |
| 4. Intention to leave studies on all categories of health workers (clinical and other service providers). | 4. Studies investigated the Intention to leave (change) current job. |
| 5. Intention to leave studies done on private or public healthcare settings | |
| 6. Studies written in the English language. | |
| 7. Methodological quality >50%. | |

downloaded and evaluated against the inclusion criteria. All selected articles from this stage were saved in a separate folder for quality assessment and those that passed the quality assessment were used for this evidence synthesis. Papers that were excluded based on the full-text assessment against the inclusion and exclusion criteria were justified. The entire screening and selection process was undertaken using Rayyan software [33] independently by two authors, who later compared their results from the software and there were no major discrepancies and minor discrepancies were resolved based on consensus between the two authors.

## Quality assessment (appraisal) of studies

A critical appraisal tool based on the study design was used to assess the methodological quality assessment of included studies. A critical appraisal checklist developed by JBI for prevalence studies has been used for this study (S1 File). Two reviewers GDA and YKG have appraised the studies independently using the abovementioned tool. The tool encompassed 9 criteria for rating different quality elements and studies scored 5 and above out of 9 were included in this meta-analysis study. During the quality assessment disagreements were solved through discussion and before an agreement reached weighted kappa index was done, which was 98.33% [32, 34].

## Data extraction

JBI Standardized data extraction template was used to ensure the extraction of the same types of data across the included studies. The two reviewers GDA and YKG extracted the entire necessary data independently using the SRDR+ online platform [35]. The data collection included the following items; study details, study method and results.

## Data synthesis and meta-analysis

Data extracted from relevant studies were analyzed using tables, graphs, and other diagrams to investigate how studies compare to each other. Meta-analysis was conducted to pool health workers prevalence of intention to leave by using random and fixed effect models. The data from all included studies were synthesized according to outcome variables and study designs using Meta Essentials v 1.5 [36, 37], R software version 4.3.1 and MedCalc v 20.114 [38] statistical tools. The extent of heterogeneity across studies was checked using the Q-test and $I^2$-test (I squared >50% indicating significant heterogeneity). To study the effect of covariates on the pooled effect size and the heterogeneity across studies, subgroup analysis was conducted on;

the sample size of individual studies, year of publication, health care setting, region and health worker categories. The effect of covariates on the pooled effect size was considered significant when the p-value was <0.05 or their 95% CI did not cross one.

## Results

### Screening and selection process of included studies

During the bibliographic data base search, a total of 2, 972 studies were identified. Of which 1, 005 were duplicates and removed. Of 1, 967 remaining studies, 1, 903 were found to be unrelated and excluded based on title and abstract screening. Finally, 64 full text articles were downloaded for further evaluation against eligibility criteria and 15 full text articles excluded as they missed prevalence of turnover intention data and 31 full text articles found unrelated for this systematic review and excluded. Four redundant full text articles were excluded. At last, we ended up with 14 full text articles for evidence synthesis (Fig 1).

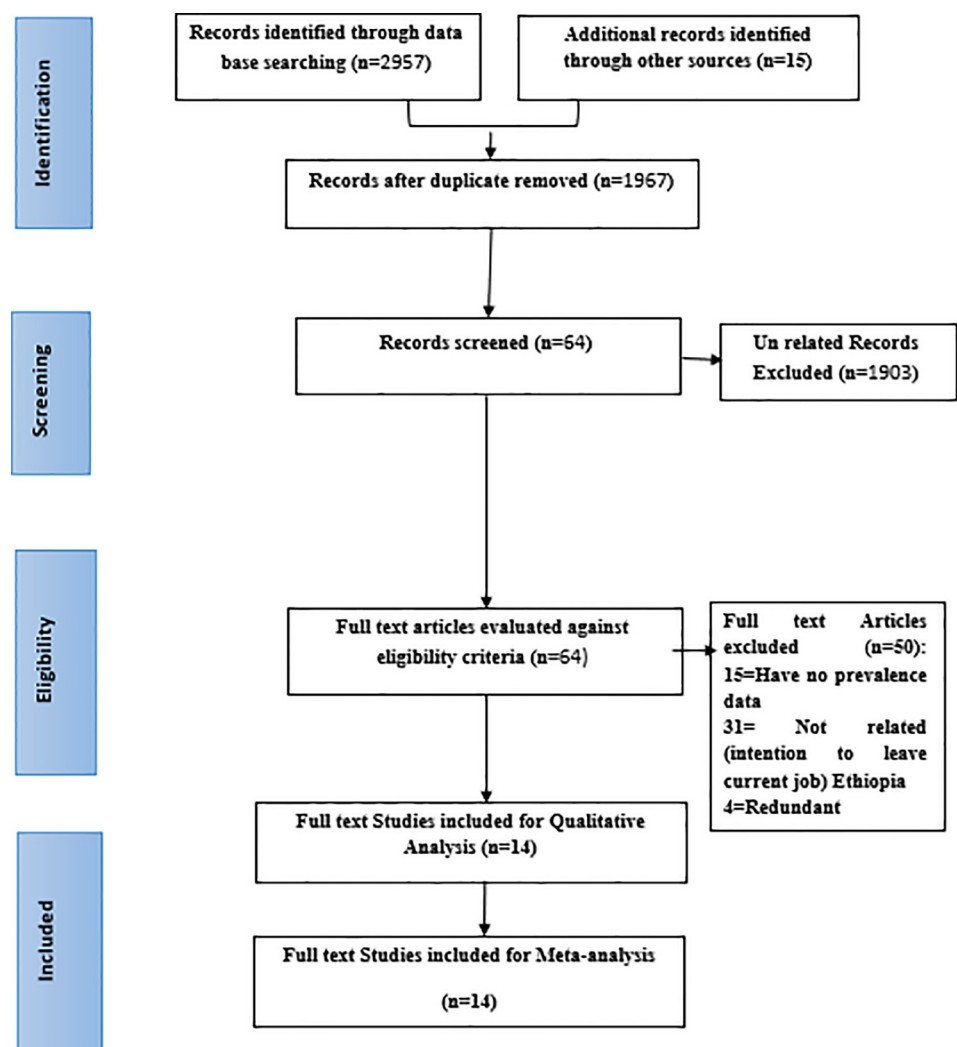

**Fig 1. Flow chart presenting the study selection with the preferred reporting items for systematic review and meta-analysis (PRISMA) guidelines on intention to leave among healthcare workers in Ethiopia, 2023.**

**Table 2. Characteristics of included studies for systematic review and meta-analysis on intention to leave their current health facility (n = 14), 2023.**

| Study ID | Publication Year | Location | Study Setting | Participants | Design | Data Type | Procedure | Sample Size | Event | Response rate | Prevalence |
|---|---|---|---|---|---|---|---|---|---|---|---|
| Adugna Endale Woldegiorgis, 2015 | 2015 | Gamb | HOS&HC | All | CS | Quan | SRS | 252 | 122 | 98.40% | 0.480 |
| Andualem Wubetie, 2020 {Citation} | 2020 | A.A | HOS | NRS | CS | Quan | Census | 102 | 79 | 91.10% | 0.775 |
| ASCHALEW MESFIN, 2022 | 2022 | Amh | HOS | All | CS | Quan | SRS | 242 | 140 | 98% | 0.580 |
| Aster Ferede, 2018 | 2018 | Amh | HOS&HC | All | CS | Quan | MSSS | 568 | 348 | 92.80% | 0.61 |
| Aynye Negesse Woldekiros, 2022 | 2022 | A.A | HOS | NRS | CS | Quant | SRS | 392 | 316 | 96.10% | 0.806 |
| Dawit Gebregziabher, 2020 | 2020 | Tig | HOS | NRS | CS | Quant | SyRS | 148 | 96 | 100% | 0.649 |
| Endager Abera, 2014 | 2014 | Amh | HOS | All | CS | Quant | StRS | 394 | 207 | 93.33% | 0.525 |
| Endalkachew Dellie, 2019 | 2019 | Amh | HOS | LAB | CS | Quant | StClS | 336 | 220 | 91.80% | 0.655 |
| Fikirte Girma, 2021 | 2021 | A.A | HOS&HC | All | CS | Quant | SRS | 402 | 284 | 100.00% | 0.707 |
| Girma Alem Getie, 2015 | 2015 | Amh | HOS&HC | NRS | CS | Quant | SRS | 372 | 221 | 87.84% | 0.594 |
| Hailay Abrha Gesesew, 2016 | 2016 | Oro | HOS&HC | All | CS | Mix | SRS | 367 | 218 | 87.00% | 0.594 |
| Hangasu Udess, 2020 | 2020 | Oro | HOS&HC | All | CS | Quan | StRS | 256 | 140 | 97.7% | 0.547 |
| Nigusu Worku, 2019 | 2019 | Amh | HOS | All | CS | Mix | SRS | 382 | 259 | 93.60% | 0.678 |
| Tilahun Mekonnen, 2022 | 2022 | Oro | HOS&HC | All | CS | Mix | LM | 362 | 235 | 100.00% | 0.65 |

Amha = Amhara, Gamb = gambela, HOS = hospital, HC = health center, MSSS = multistage stratified sampling, SRS = Simple random sampling, SyRS = systematic random sampling, StRS = stratified random sampling, LAB = laboratory, StClS = stratified cluster sampling, Oro = oromia, Mix = mixed, Tig = Tigray, LM = lottery method

## Characteristics of included studies

Among 14 included studies published from 2013–2022 in Ethiopia, the smallest sample size was 102 on a study done in Addis Ababa and the largest one was 568 on a study done in Amhara regional state. The study settings included were health centers and hospitals. These studies involved a total of 4575 participants (Table 2).

## Methodological quality of included studies

Methodological quality assessment was done using the JBI's Prevalence Studies Critical Appraisal Checklist. Assessment was done by two independent assessors, GDA and YKG. Discrepancies between assessors were resolved through consensus and mutual understanding. The assessment results revealed that quality of included studies range from 66.7% to 100% (S2 File).

## Pooled prevalence of intention to leave

Based on 14 included studies with 4575 participants, the pooled intention to leave their current health facilities among health workers at Ethiopian healthcare setting was found to be 63.25% (95% CI (61.84–64.65)) for fixed effect model and 63.52% (95% CI (58.606–67.904)) for Random effect model at Q = 141.57 ($I^2$(inconsistency) = 90.82%, P < 0.0001). Only the results from random effect model were used for further discussion as there was significant heterogeneity between studies (Table 3 and Fig 2).

**Table 3. Prevalence of health worker Intention to leave at health care settings in Ethiopia (n = 14), 2023.**

| Study ID | Sample size | Proportion (%) | 95% CI | Weight (%) | |
|---|---|---|---|---|---|
| | | | | Fixed | Random |
| Adugna Endale Woldegiorgis, 2015 | 252 | 48.41 | 42.10 to 54.77 | 5.51 | 7.06 |
| Andualem Wubetie, 2020 | 102 | 77.45 | 68.11 to 85.14 | 2.24 | 6.05 |
| ASCHALEW MESFIN, 2022 | 242 | 57.85 | 51.36 to 64.15 | 5.30 | 7.03 |
| Aster Ferede, 2018 | 568 | 61.27 | 57.12 to 65.29 | 12.40 | 7.54 |
| Aynye Negesse Woldekiros, 2022 | 392 | 80.61 | 76.35 to 84.41 | 8.56 | 7.36 |
| Dawit Gebregziabher, 2020 | 148 | 64.87 | 56.60 to 72.52 | 3.25 | 6.54 |
| Endager Abera, 2014 | 394 | 52.54 | 47.48 to 57.56 | 8.61 | 7.37 |
| Endalkachew Dellie, 2019 | 336 | 65.48 | 60.12 to 70.55 | 7.34 | 7.27 |
| Fikirte Girma, 2021 | 402 | 70.65 | 65.93 to 75.06 | 8.78 | 7.38 |
| Girma Alem Getie, 2015 | 372 | 59.41 | 54.23 to 64.44 | 8.13 | 7.33 |
| Hailay Abrha Gesesew, 2016 | 367 | 59.40 | 54.18 to 64.47 | 8.02 | 7.33 |
| Hangasu Udess, 2020 | 256 | 54.69 | 48.37 to 60.90 | 5.60 | 7.07 |
| Nigusu Worku, 2019 | 382 | 67.80 | 62.86 to 72.46 | 8.35 | 7.35 |
| Tilahun Mekonnen, 2022 | 362 | 64.92 | 59.76 to 69.83 | 7.91 | 7.32 |
| Total (fixed effects) | 4575 | 63.25 | 61.84 to 64.65 | 100.00 | 100.00 |
| Total (random effects) | 4575 | 63.52 | 58.61 to 67.90 | 100.00 | 100.00 |

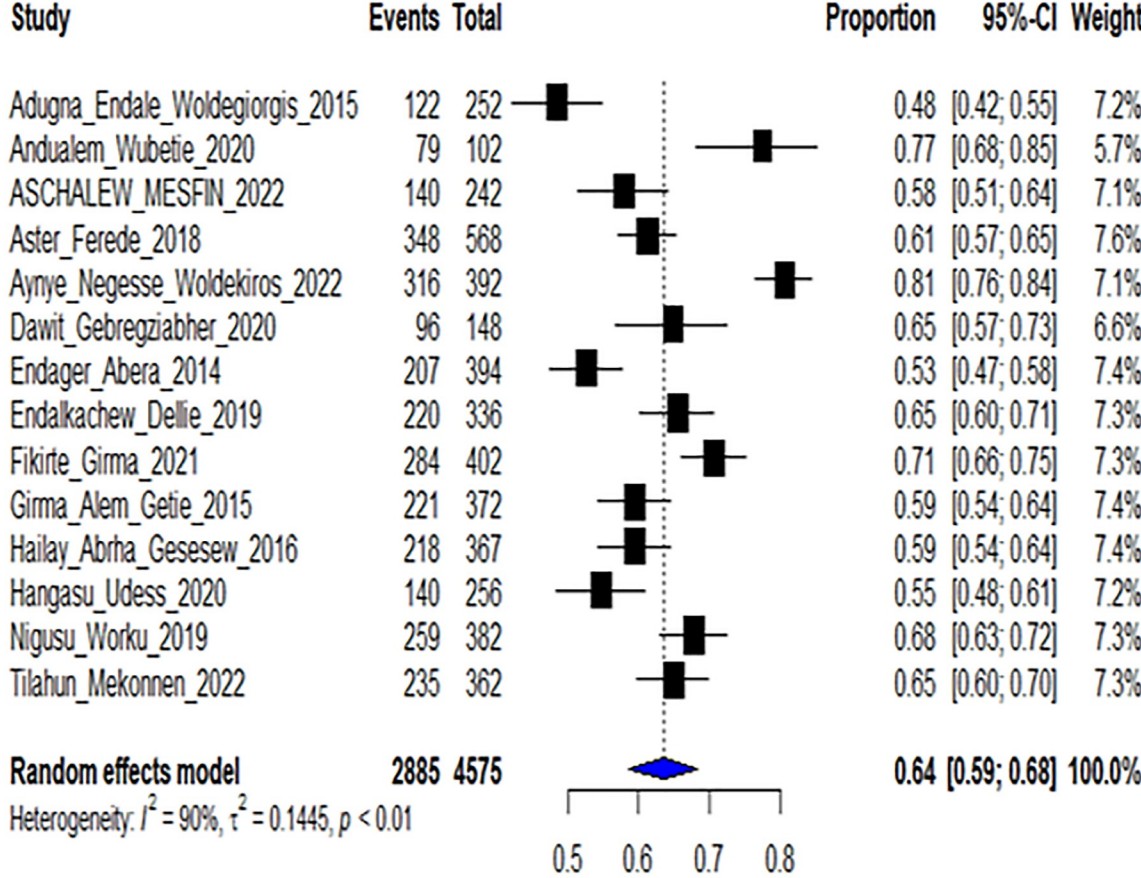

**Fig 2. Forest plot showing the effect size of intention to leave the current health facility among health workers in Ethiopia, 2023.**

**Table 4. Egger's and Begg's test for publication bias.**

| Egger's test | |
|---|---|
| Intercept | 0.02467 |
| 95% CI | -10.6577 to 10.7071 |
| Significance level | P = 0.9961 |
| **Begg's test** | |
| Kendall's Tau | -0.1209 |
| Significance level | P = 0.5470 |

## Publication bias analysis of included studies

The Egger's test and symmetric funnel plot showed that studies included in this meta-analysis have no publication bias (Table 4 and Fig 3).

## Subgroup analysis of included studies

To investigate source of heterogeneity, sub group analysis has been done on sample size, study area, participant's profession, year of publication and health facility type. All the above listed variables were significant source of statistical heterogeneity of which year of publication was the highest source of heterogeneity ($I^2$ = 99.02%, Table 5).

## Factors associated with health worker's intention to leave at health care settings in Ethiopia

Fourteen included studies used for pooled prevalence of intention to leave and analysis of factors associated with it. Factors assessed were; autonomy, co- worker relationship, job satisfaction, organizational justice, payment and benefits system, performance appraisal, recognition, training and development opportunity, and workload. Of which only organizational justice OR = 0.29(0.14–0.61) was found to be significantly associated factors for intention to leave of health workers at healthcare setting in Ethiopia (Table 6).

## Discussion

The aim of this systematic review and meta-analysis was to assess the related factors and pooled prevalence of intention to leave their current health facilities among health workers in Ethiopia. We studied intention to leave as a direct predictor and proactive approach to determine the pooled prevalence of turnover among health workers in Ethiopia.

The distribution of included 14 studies throughout Ethiopia was 21.4%, 42.9%, 7.1%, 21.4% and 7.1% in Addis Ababa, Amhara, Gambela, Oromia and Tigray respectively. The smallest sample size among included studies was 102 and the largest one was 568 from studies done in Addis Ababa and Amhara respectively. The response rate of studies ranges from 87% to 100% and publication year was from 2015 to 2022.

From this review, we synthesized that among 14 individual studies, the pooled intention to leave their current health facility among health workers in Ethiopia was 63.52% (95% CI (58.61–67.90)) for Random effect model at Q = 141.57 ($I^2$ (inconsistency) = 90.82%, P < 0.0001). We used only the results from the random effect model as there was significant heterogeneity between studies. The highest prevalence of intention to leave was 80.6% and the lowest was 48.4% as reported by a study done in Addis Ababa and Gambela, in 2022 and 2015 respectively.

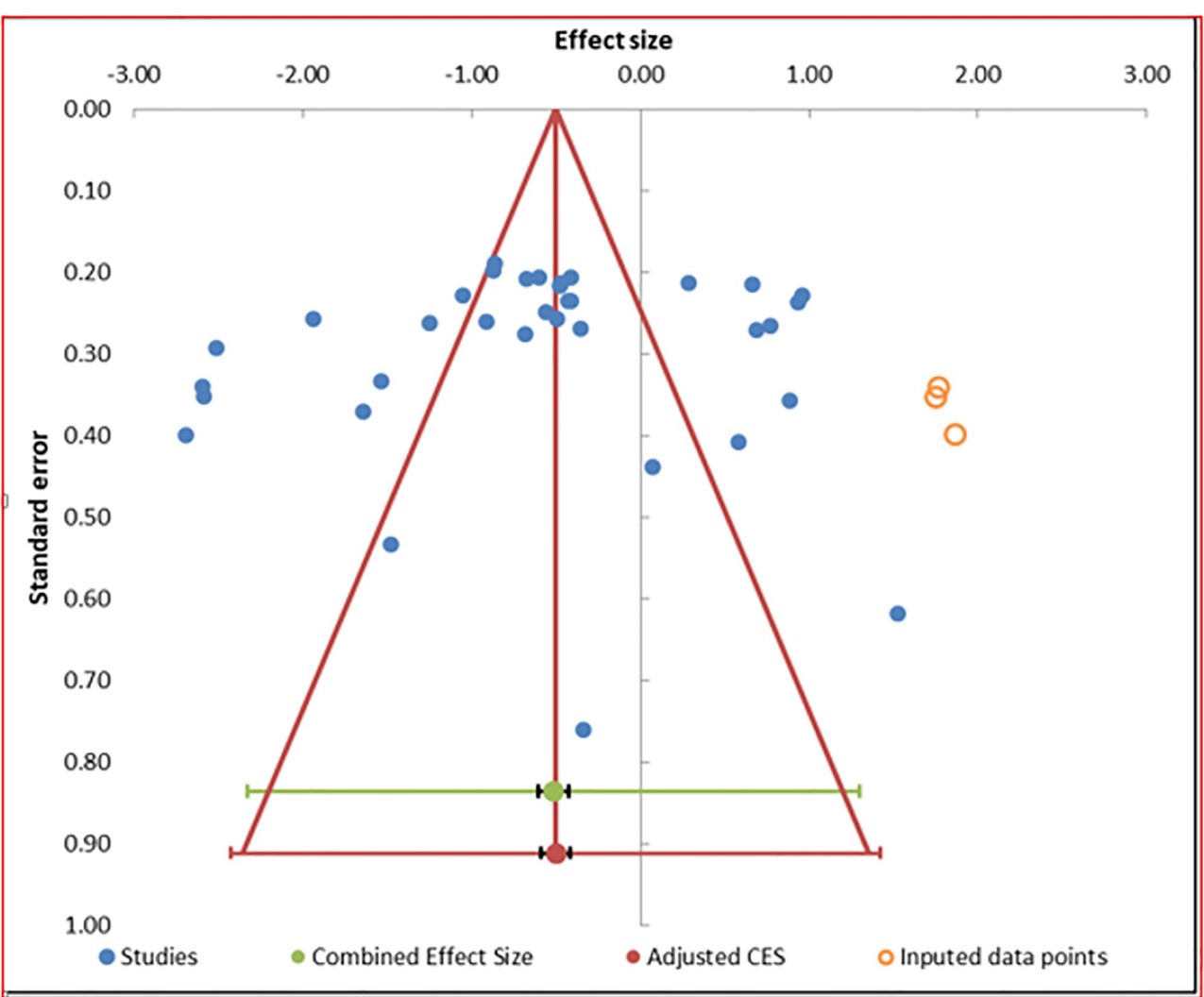

**Fig 3. Funnel plot showing the symmetry of included articles.**

This finding was higher than a study done on general practitioners in 2020 globally which was reported as 47% [23]. The difference may be due to variations in characteristics of study participants, socio economic status countries and variations in health work force retention mechanisms.

Our finding is higher than a cross sectional study of New Jersey hospital nurses intention to leave their hospital (36.5%) [39]. This difference may be due to variations of study design and study participants, all health workers versus nurses. When compared with a report of 30.4% intention to leave Chinese primary health workers by a systematic review and meta-analysis done in 2020, our finding was by far higher [40]. The difference may be due to variations in characteristics of study participants, socio economic status of countries and variations in health work force retention mechanisms.

This finding is higher when compared to African countries like Tanzania (18.8%), Malawi (26.5%) and South Africa (41.4%) [26]. This difference may be due to variations with respect to study procedures, participants and study settings. Significant heterogeneity was observed in

**Table 5. Subgroup analysis of included studies for systematic review and meta-analysis on intention to leave their current health facility, 2023.**

| Subgroup Name | | Odds Ratio | 95% CI | Weight |
|---|---|---|---|---|
| **Location**<br>Q = 113.48, $T^2$ = 0.22, PQ = 0.001, $I^2$ = 96.48%, 95%CL Pooled Odds Ratio = 0.97 (0.52, 1.81), Weighted Average = 12.80% | Addis Ababa | 2.09 | 1.77, 2.47 | 20.45% |
| | Amhara | 0.82 | 0.73, 0.93 | 20.76% |
| | Gambela | 0.53 | 0.41, 0.68 | 19.62% |
| | Oromia | 0.86 | 0.74, 0.99 | 20.61% |
| | Tigray | 1.08 | 0.77, 1.52 | 18.56% |
| **Participants**<br>Q = 58.55, $T^2$ = 0.20, PQ = 0.001, $I^2$ = 96.58%, 95%CL Pooled Odds Ratio = 1.05(0.39, 2.82), Weighted Average = 12.32% | All | 0.69 | 0.60, 0.79 | 33.88% |
| | Lab | 1.12 | 0.89, 1.41 | 32.42% |
| | Nurse | 1.51 | 1.30, 1.75 | 33.70% |
| **Publication Year**<br>Q = 101.93, $T^2$ = 0.39, PQ = 0.001, $I^2$ = 99.02%, 95%CL Pooled Odds Ratio = 1.00(0.00, 273.88), Weighted Average = 3.33% | 2014–2018 | 0.64 | 0.57, 0.73 | 50.00% |
| | 2019–2022 | 1.56 | 1.38, 1.76 | 50.00% |
| **Response Rate**<br>Q = 10.80, $T^2$ = 0.04, PQ = 0.001, $I^2$ = 90.74%, 95%CL Pooled Odds Ratio = 1.00 (0.16, 6.19), Weighted Average = 31.53% | <95 | 0.87 | 0.77, 0.98 | 50.00% |
| | > = 95 | 1.15 | 1.02, 1.30 | 50.00% |
| **Sample Size**<br>Q = 55.41, $T^2$ = 0.11, PQ = 0.001, $I^2$ = 92.78%, 95%CL Pooled Odds Ratio = 1.05 (0.68, 1.61), Weighted Average = 27.04% | 100–200 | 1.39 | 1.05, 1.84 | 18.42% |
| | 201–300 | 0.62 | 0.53, 0.73 | 20.64% |
| | 301–400 | 1.14 | 1.01, 1.28 | 21.15% |
| | 401–500 | 1.45 | 1.16, 1.82 | 19.51% |
| | 501–600 | 0.92 | 0.77, 1.10 | 20.28% |
| **Study Setting**<br>Q = 25.92, $T^2$ = 0.10, PQ = 0.001, $I^2$ = 96.14%, 95%CL Pooled Odds Ratio = 1.01 (0.06, 17.14), Weighted Average = 12.99% | Hospital | 1.25 | 1.11, 1.41 | 50.00% |
| | Hosp and HC | 0.80 | 0.71, 0.90 | 50.00% |

regional states on intention to leave as 64.86% in Tigray regional state, 60.81% in Amhara regional state, 75.78% in Addis Ababa city administration and 60.20% in Oromia regional state. This difference may be due to variations in organizational justice and leadership style of health facilities.

Prevalence of intention to leave in Ethiopia at healthcare settings was 65.98% and 60.80% at hospitals only and, studies done at both hospitals and health centers levels respectively. This may be due to higher workloads pressures at hospitals. Based on health worker categories, the intention to leave was 60.56%, 65.48% and 70.22% among studies that include all health categories, studies of laboratories only and studies of nurses only respectively. This difference may be due to variations in workload, incentive package and other professional specific determinants.

The relation between Intention to leave the current health facility and associated factors have been analyzed. The factors undergone statistical test were; professional; autonomy, co-worker relationship, job satisfaction, organizational justice, payment and benefits system, performance appraisal, recognition, training and development opportunity and workload. Organizational justice here refers to the fairness of (resource) allocation within the organization. This encompasses the actions and decisions made by management, employees' salary, opportunities for promotion and advancement, performance evaluation, etc. Employees expect the facility to be just, unbiased, and transparent about these aspects. If everything follows the proper course and the employees are treated fairly, they tend to be motivated and satisfied. Lack of organizational justice can demotivate employees and make them leave the organization [41]. Likewise, our finding should that organizational justice was found to have a significant association with intention to leave their current health facility as pooled odds ratio 0.29 at 95% CL (0.14–0.61). The odds of intention to leave among healthcare workers who believe there is

**Table 6. Factors associated with health worker's intention to leave their current health care settings in Ethiopia, 2023.**

| Factors | Study ID | Odds Ratio | CI | Weight | Pooled Odds Ratio | Q | I² | T² |
|---|---|---|---|---|---|---|---|---|
| Autonomy | Girma Alem Getie, 2015 | 1.34 | 0.88–2.03 | 6.53% | 1.26(0.09–16.83) | 17.96 | 88.86% | 0.61 |
| | Tilahun Mekonnen, 2022 | 0.51 | 0.34–0.76 | | | | | |
| | Wubetie et al. BMC Nursing (2020) | 4.59 | 1.35–15.64 | | | | | |
| Co- worker relationship | Aynye Negesse Woldekiros, 2022 | 0.23 | 0.08–0.65 | 7.15% | 0.63(0.05–7.37) | 10.60 | 81.13% | 0.58 |
| | Tilahun Mekonnen, 2022 | 0.55 | 0.36–0.82 | | | | | |
| | Wubetie et al. BMC Nursing (2020) | 1.79 | 0.79–4.01 | | | | | |
| Job Satisfaction | Adugna Endale Woldegiorgis, 2015 | 0.07 | 0.03–0.15 | 11.91% | 0.63(0.23–1.77) | 82.59 | 92.74% | 0.68 |
| | ASCHALEW MESFIN, 2022 | 2.16 | 1.28–3.64 | | | | | |
| | Aster Ferede, 2018 | 0.42 | 0.29–0.61 | | | | | |
| | Endalkachew Dellie, 2019 | 0.66 | 0.42–1.05 | | | | | |
| | Girma Alem Getie, 2015 | 1.94 | 1.27–2.95 | | | | | |
| | Hailay Abrha Gesesew, 2014 | 0.62 | 0.40–0.95 | | | | | |
| | Tilahun Mekonnen, 2022 | 0.66 | 0.44–0.99 | | | | | |
| Organizational Policy | ASCHALEW MESFIN, 2022 | 0.40 | 0.24–0.67 | 22.58% | **0.29(0.14–0.61)** * | 14.94 | 73.22% | 0.23 |
| | Tilahun Mekonnen, 2022 | 0.42 | 0.28–0.62 | | | | | |
| | Endalkachew Dellie, 2019 | 0.14 | 0.09–0.24 | | | | | |
| | Fikirte Girma,2021 | 0.19 | 0.09–0.40 | | | | | |
| | Wubetie et al. BMC Nursing (2020) | 0.71 | 0.16–3.21 | | | | | |
| Payment and benefits system | Adugna Endale Woldegiorgis, 2015 | 0.07 | 0.04–0.15 | 5.48% | 0.34(0.05–2.13) | 75.41 | 94.70% | 1.73 |
| | Aynye Negesse Woldekiros, 2022 | 0.61 | 0.37–1.01 | | | | | |
| | Endalkachew Dellie, 2019 | 0.08 | 0.04–0.15 | | | | | |
| | Gebregziabher et al. BMC Nursing (2020) | 2.43 | 1.20–4.92 | | | | | |
| | Hangasu Udess, 2020 | 0.50 | 0.29–0.87 | | | | | |
| Performance Appraisal | Endalkachew Dellie, 2019 | 0.57 | 0.35–0.93 | 24.23% | 0.44(0.02–10.11) | 2.14 | 53.20% | 0.06 |
| | Nigusu Worku, 2019 | 0.35 | 0.22–0.55 | | | | | |
| Recognition | Endalkachew Dellie, 2019 | 0.29 | 0.17–0.48 | 9.08% | 0.54(0.11–2.60) | 36.99 | 91.89% | 0.83 |
| | Nigusu Worku, 2019 | 0.65 | 0.41–1.03 | | | | | |
| | Fikirte Girma,2021 | 0.21 | 0.11–0.41 | | | | | |
| Training and development opportunity | Endalkachew Dellie, 2019 | 0.08 | 0.05–0.14 | 2.37% | 0.61(0.01–54.29) | 87.85 | 97.72% | 4.03 |
| | Gebregziabher et al. BMC Nursing (2020) | 1.07 | 0.45–2.56 | | | | | |
| | Girma Alem Getie, 2015 | 2.60 | 1.66–4.08 | | | | | |
| Workload | Aynye Negesse Woldekiros, 2022 | 0.70 | 0.41–1.19 | 10.67% | 1.04(0.15–7.20) | 21.99 | 90.91% | 0.57 |
| | Endalkachew Dellie, 2019 | 2.54 | 1.60–4.06 | | | | | |
| | Tilahun Mekonnen, 2022 | 0.63 | 0.41–0.96 | | | | | |

* = p<0.05

organizational justice was 0.71 times less than that of healthcare workers who believe there was no organizational justice. Comprehensive global studies on public sector employee turnover intension supports this finding as organizational justice like participation in decision making and transparency are predictors of employees' intention to leave their facility [42].

## Limitation

This systematic review and meta-analysis have some limitations. First, the included studies were heterogeneous with respect to study procedures, participants and study settings. Accordingly, our findings were summarized on a broader level, which inherently suppresses some of

the unique features of different approaches. Second, the majority of studies were based on a cross-sectional study design that limits the precision of intention to leave results. Third, the included articles were only full-text articles freely accessible which compromises the quality and generalizability of the review.

## Conclusion

From this review, we synthesized that the prevalence of pooled intention to leave healthcare settings among health workers in Ethiopia was more than 6 in 10 health workers. This result was higher than studies done in other parts of the world, even the African countries. This discrepancy might be due to variations in socio-economic, political and studied health worker categories. Based on this systematic review and meta-analysis, the associated factor for health workers' intention to leave their current health facility was only organizational justice.

## Recommendation

Health facilities in Ethiopia should strive to have organizational justice like transparent actions and decisions made by management, employees' salary, opportunities for promotion and advancement, etc. to retain health workers. To be more specific and precise, the authors recommend further systematic review and meta-analysis on a departmental basis and conduct an original study on administrative staff who are working in finance, human resources and health professionals who are not included in this study.

## Supporting information

**S1 File. Quality assessment (appraisal) tool.**
(DOCX)

**S2 File. Methodological quality of included studies.**
(DOCX)

**S1 Checklist. PRISMA checklist.**
(DOCX)

## Acknowledgments

It is a great pleasure to thank Yirga Yeshiwas Alefe, CEO of Bure Asrade Zewude Hospital for frequent consultation on how to conduct a systematic review. The authors would like to acknowledge the contributions made by MedCalc for 14-day free trial meta-analysis database and freely accessed Meta essentials database for meta-analysis. We would also like to acknowledge JBI for the live webinar by Associate Professor Edoardo Aromataris on 'How to conduct systematic review: Latest guidance' and Campbell collaborative on how to calculate effect size for meta-analysis by Prof. David Wilson.

## Author Contributions

**Conceptualization:** Gizew Dessie Asres, Yeshiwork Kebede Gessesse.

**Data curation:** Gizew Dessie Asres, Yeshiwork Kebede Gessesse.

**Formal analysis:** Gizew Dessie Asres, Yeshiwork Kebede Gessesse.

**Investigation:** Gizew Dessie Asres, Yeshiwork Kebede Gessesse.

**Methodology:** Gizew Dessie Asres, Yeshiwork Kebede Gessesse.

**Project administration:** Gizew Dessie Asres, Yeshiwork Kebede Gessesse.

**Software:** Gizew Dessie Asres.

**Visualization:** Yeshiwork Kebede Gessesse, Molalign Tarekegn Minwagaw.

**Writing – original draft:** Gizew Dessie Asres.

**Writing – review & editing:** Gizew Dessie Asres, Yeshiwork Kebede Gessesse, Molalign Tarekegn Minwagaw.

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
