## [Decision Letter · Decision Letter 0]

25 Jan 2024

PGPH-D-23-02236

Intention to Leave the Current Health Facility among Healthcare Workers in Ethiopia: Systematic Review and Meta-analysis

Dear Dr. Asres,

Thank you for submitting your manuscript to PLOS Global Public Health. After careful consideration, we feel that it has merit but does not fully meet PLOS Global Public Health’s publication criteria as it currently stands. Therefore, we invite you to submit a revised version of the manuscript that addresses the points raised during the review process.

Please see the comments of two reviewers below and in the attachment. Note that both reviewers have queried what can be said about the reasons for leaving, which would be useful to clarify. Both reviewers have also pointed out that the discussion and conclusions could be broader in scope, and that some copyediting is required.

We look forward to receiving your revised manuscript.

Kind regards,

Hanna Landenmark

Staff Editor

Journal Requirements:

1. We noticed you have some minor occurrence of overlapping text with the following previous publication(s), which needs to be addressed:

- https://www.britishjournalofnursing.com/content/literature-review/factors-influencing-retention-among-hospital-nurses-systematic-review

2. In your revision ensure you cite all your sources (including your own works), and quote or rephrase any duplicated text outside the methods section. Further consideration is dependent on these concerns being addressed.

3. Please provide separate figure files in .tif or .eps format only and remove any figures embedded in your manuscript file. Please also ensure all files are under our size limit of 10MB.

4. In the online submission form, you indicated that "All necessary information was attached to this manuscript. Whenever an additional data set is needed, Data sharing is applicable upon reasonable request to the corresponding author". All PLOS journals now require all data underlying the findings described in their manuscript to be freely available to other researchers, either 1. In a public repository, 2. Within the manuscript itself, or 3. Uploaded as supplementary information.

Additional Editor Comments (if provided):

Reviewers' comments:

Reviewer's Responses to Questions

**Comments to the Author**

1. Does this manuscript meet PLOS Global Public Health’s publication criteria? Is the manuscript technically sound, and do the data support the conclusions? The manuscript must describe methodologically and ethically rigorous research with conclusions that are appropriately drawn based on the data presented.

Reviewer #1: Yes

Reviewer #2: Partly

2. Has the statistical analysis been performed appropriately and rigorously?

Reviewer #1: I don't know

Reviewer #2: Yes

3. Have the authors made all data underlying the findings in their manuscript fully available (please refer to the Data Availability Statement at the start of the manuscript PDF file)?

Reviewer #1: Yes

Reviewer #2: Yes

4. Is the manuscript presented in an intelligible fashion and written in standard English?

Reviewer #1: Yes

Reviewer #2: No

5. Review Comments to the Author

Reviewer #1: Thank you for addressing the significant issue of intention to leave healthcare. I have two major concerns detailed below followed by minor suggestions.

• Can the authors elaborate on the factors associated with the intention to leave? Currently only a short paragraph is dedicated to this very important aspect of the manuscript and an underdeveloped section that would greatly improve the paper if more details were provided. What aspects of organizational policy were found to be significant ? How was organizational policy defined across all studies? The term organizational policy is a very broad category and does not provide much information to the reader. Why were these factors the only factors assessed? Further can the authors break these results down by type of worker? Factors impacting nurses may be different from those working in the pharmacy or in food services. Also, Table 6 is missing

• The Discussion focuses primarily on repeating the results – there is very little explanation as to why the intent to leave is relatively high in Ethiopia and what could be done to address these issues. Further the authors only attribute differences with their study and others to study procedures, participants, and the study setting - what other reasons could be associated with intent to leave and how do they differ or are they similar with the current study? The Discussion section needs more development

• The paper overall is limited by inaccuracies and sentence structure, grammar, and wording.

• There are several statements highlighted in the attached that require accurate references.

• Some of the references are not accurately formatted

• Acronyms need to be defined prior to their first use

• Tables 2 is missing

Additional comments can be found in the attachment

Reviewer #2: Dear Authors,

I read with great interest the manuscript received for evaluation. Your work investigates an important and current health issue. I have several comments aimed at enhancing its quality.

1. In Ethiopia, there is no representative and summarized data on health professionals’ intention to

leave their current health facility. Some studies on intention to leave did not clearly state whether

it is the intention to leave their current profession or their current working facility. The paucity of

previous studies showed inconsistent prevalence that ranges from 30.4 % to 80.6% (26,27).

Therefore, this systematic review and meta-analysis aimed to estimate the pooled prevalence of

health professionals’ intention to leave their current health facility and factors associated with it

in Ethiopia. But, there is a systematic review and meta-analysis conducted in Ethiopia on this topic. What distinguishes this work from the existing literature/recently published systematic review and meta analysis done in Ethiopia?

2. Your study conclusion needs more clarifications.

3. The authors should revise the language to improve readability

6. PLOS authors have the option to publish the peer review history of their article (what does this mean?). If published, this will include your full peer review and any attached files.

**Do you want your identity to be public for this peer review?** For information about this choice, including consent withdrawal, please see our Privacy Policy.

Reviewer #1: No

Reviewer #2: No

---

## [Author Response · Author response to Decision Letter 0]

23 Feb 2024

Reviewer 1 Comments to the Author

Thank you for addressing the significant issue of intention to leave healthcare. I have two major concerns detailed below followed by minor suggestions.

• Can the authors elaborate on the factors associated with the intention to leave? Currently only a short paragraph is dedicated to this very important aspect of the manuscript and an underdeveloped section that would greatly improve the paper if more details were provided. What aspects of organizational policy were found to be significant? How was organizational policy defined across all studies? The term organizational policy is a very broad category and does not provide much information to the reader. Why were these factors the only factors assessed? Further can the authors break these results down by type of worker? Factors impacting nurses may be different from those working in the pharmacy or in food services. Also, Table 6 is missing

• The Discussion focuses primarily on repeating the results – there is very little explanation as to why the intent to leave is relatively high in Ethiopia and what could be done to address these issues. Further the authors only attribute differences with their study and others to study procedures, participants, and the study setting - what other reasons could be associated with intent to leave and how do they differ or are they similar with the current study? The Discussion section needs more development

• The paper overall is limited by inaccuracies and sentence structure, grammar, and wording.

• There are several statements highlighted in the attached that require accurate references.

• Some of the references are not accurately formatted

• Acronyms need to be defined prior to their first use

• Tables 2 is missing

Additional comments can be found in the attachment

Author response to reviewer 1 comments: Dear reviewer thank you for your line-by-line review and detailed comments to making the manuscript more enriched so that readers can get sufficient information. We made corrections per your recommendation and the changes are indicated by track changes from the original manuscript. Here is the point-by-point response to all raised queries by locating where these queries are addressed from the revised manuscript.

Reviewer question 1: Can the authors elaborate on the factors associated with the intention to leave? Currently only a short paragraph is dedicated to this very important aspect of the manuscript and an underdeveloped section that would greatly improve the paper if more details were provided. What aspects of organizational policy were found to be significant? How was organizational policy defined across all studies? The term organizational policy is a very broad category and does not provide much information to the reader. Why were these factors the only factors assessed? Further can the authors break these results down by type of worker? Factors impacting nurses may be different from those working in the pharmacy or in food services. Also, Table 6 is missing

Authors response to reviewer’s question 1: Thank you for this critical question. Based on accessed literatures, the authors identified possible factors for intention to leave their current facility as; autonomy, co- worker relationship, job satisfaction, organizational policy, payment and benefits system, performance appraisal, recognition, training and development opportunity and workload. But when statistically analyzed, only organizational policy was found to have significant relationship with the health workers intention to leave their current health facility. Organizational policy here considered as organizational justice which refers to the fairness of (resource) allocation within the organization. This encompasses the actions and decisions made by management, employees’ salary, opportunities for promotion and advancement, performance evaluation, etc. Employees expect the facility to be just, unbiased, and transparent about these aspects. If everything follows the proper course and the employees are treated fairly, they tend to be motivated and satisfied. Lack of organizational justice can demotivate employees and make them leave the organization. Those health workers who believe their facility has organizational justice tend to have lower intention to leave their current health facilities as indicated at table 6. Authors try to break the results by type of participant as indicated from table 5.

Reviewer question 2: The Discussion focuses primarily on repeating the results – there is very little explanation as to why the intent to leave is relatively high in Ethiopia and what could be done to address these issues. Further the authors only attribute differences with their study and others to study procedures, participants, and the study setting - what other reasons could be associated with intent to leave and how do they differ or are they similar with the current study? The Discussion section needs more development

• The paper overall is limited by inaccuracies and sentence structure, grammar, and wording.

• There are several statements highlighted in the attached that require accurate references.

• Some of the references are not accurately formatted

• Acronyms need to be defined prior to their first use

• Tables 2 is missing

Additional comments can be found in the attachment

Authors response to reviewer’s question 2: Thank you for your constructive and suggestive comments. All your comments are addressed on their respective sections with track changes on revised manuscript.

Reviewer 2 Comments to the Author

Dear Authors,

I read with great interest the manuscript received for evaluation. Your work investigates an important and current health issue. I have several comments aimed at enhancing its quality.

1. In Ethiopia, there is no representative and summarized data on health professionals’ intention to leave their current health facility. Some studies on intention to leave did not clearly state whether

it is the intention to leave their current profession or their current working facility. The paucity of

previous studies showed inconsistent prevalence that ranges from 30.4 % to 80.6% (26,27). Therefore, this systematic review and meta-analysis aimed to estimate the pooled prevalence of

health professionals’ intention to leave their current health facility and factors associated with it

in Ethiopia. But there is a systematic review and meta-analysis conducted in Ethiopia on this topic. What distinguishes this work from the existing literature/recently published systematic review and meta-analysis done in Ethiopia?

2. Your study conclusion needs more clarification.

3. The authors should revise the language to improve readability 

Author response to reviewer 2 comments: We appreciate your concern about the similarity of the recently published systematic review on intention to leave. But our study is specifically on the intention to leave the current working health facility, which does not include intention to leave or change their profession within a health facility. 

Editor’s instruction and authors’ response to them

1. A rebuttal letter that responds to each point raised by the editor and reviewer(s). You should upload this letter as a separate file labeled 'Response to Reviewers'. 

Author Response to Editors suggestion: Dear Editor thank you for your guiding suggestion on how to address reviewers’ comments. We did write a response to reviewers per your recommendation and the changes made were indicated by track changes at the original manuscript.

Author Response to Editors suggestion: We did prepare the manuscript per your recommendation.

Author Response to Editors suggestion:3 Thank you for your guiding suggestion and we did it according to your suggestion.

---

## [Decision Letter · Decision Letter 1]

1 Jun 2024

PGPH-D-23-02236R1

Intention to Leave the Current Health Facility among Healthcare Workers in Ethiopia: Systematic Review and Meta-analysis

Dear Dr. Asres,

Thank you for submitting your manuscript to PLOS Global Public Health. After careful consideration, we feel that it has merit but does not fully meet PLOS Global Public Health’s publication criteria as it currently stands. Therefore, we invite you to submit a revised version of the manuscript that addresses the points raised during the review process.

We look forward to receiving your revised manuscript.

Kind regards,

Vanessa Carels

Staff Editor

Journal Requirements:

Additional Editor Comments (if provided):

Reviewers' comments:

Reviewer's Responses to Questions

**Comments to the Author**

1. If the authors have adequately addressed your comments raised in a previous round of review and you feel that this manuscript is now acceptable for publication, you may indicate that here to bypass the “Comments to the Author” section, enter your conflict of interest statement in the “Confidential to Editor” section, and submit your "Accept" recommendation.

Reviewer #3: All comments have been addressed

2. Does this manuscript meet PLOS Global Public Health’s publication criteria? Is the manuscript technically sound, and do the data support the conclusions? The manuscript must describe methodologically and ethically rigorous research with conclusions that are appropriately drawn based on the data presented.

Reviewer #3: Yes

3. Has the statistical analysis been performed appropriately and rigorously?

Reviewer #3: Yes

4. Have the authors made all data underlying the findings in their manuscript fully available (please refer to the Data Availability Statement at the start of the manuscript PDF file)?

Reviewer #3: Yes

5. Is the manuscript presented in an intelligible fashion and written in standard English?

Reviewer #3: Yes

6. Review Comments to the Author

Reviewer #3: This is a well-written and properly conducted systematic review and meta-analysis. Its topic is timely and relevant and fits well with the journal’s profile.

My comments and suggestions for the authors are as follows.

(1) Two important recent articles are not included in this study, at least I could not find any reference to them in the manuscript.

- Gebrekidan, A. Y. et al. (2023): https://bmjopen.bmj.com/content/bmjopen/13/5/e067266.full.pdf. In this one, the authors analysed a larger sample of articles, all about Ethiopia, all published after 2013.

- Zenebe, G. A. et al. (2024): https://www.ajol.info/index.php/emj/article/view/268493/253355. The main aim of this article is to examine health professionals’ intention to leave their jobs in public health facilities and its determinants in Ethiopia, using systematic review and meta-analysis.

The aim, structure, methods, and some of the results of the above articles are very similar to this study. For this reason, there is a risk that the novelty of the manuscript will be questioned. I therefore recommend that you include these two articles in the Introduction and Discussion of your paper, emphasizing the originality of your own work and its novelty compared to them.

(2) This study aims to assess the pooled prevalence of intention to leave their current health facility among health workers in Ethiopia. It is worth describing how pooled prevalence of intention to leave was calculated. For example, in the study of Zenebe, G. A. et al. (2024), this indicator “was calculated by dividing the total number of health professionals who intend to leave by the total sample size (number of health professionals included in the study) and multiplying by one hundred (100)”.

(3) During the study selection process, did you look for additional sources in the references of the articles already collected? I did not find any sign of this in the Methods section. If you did not use this ‘snowball’ technique, please explain why you did so.

(4) You wrote that you had collected articles and grey literature published between 2013 and 2022. Could you please explain why 2013 was chosen as the starting year?

(5) Some minor spelling and grammar issues should be revised (e.g. instead of “professional; autonomy”, did you want to write “professional autonomy”?).

7. PLOS authors have the option to publish the peer review history of their article (what does this mean?). If published, this will include your full peer review and any attached files.

**Do you want your identity to be public for this peer review?** For information about this choice, including consent withdrawal, please see our Privacy Policy.

Reviewer #3: No

---

## [Decision Letter · Decision Letter 2]

10 Jul 2024

Intention to Leave the Current Health Facility among Healthcare Workers in Ethiopia: Systematic Review and Meta-analysis

PGPH-D-23-02236R2

Dear Mr Asres,

We are pleased to inform you that your manuscript 'Intention to Leave the Current Health Facility among Healthcare Workers in Ethiopia: Systematic Review and Meta-analysis' has been provisionally accepted for publication in PLOS Global Public Health.

Best regards,

Julia Robinson

Executive Editor

Reviewer Comments (if any, and for reference):

Reviewer's Responses to Questions

**Comments to the Author**

1. If the authors have adequately addressed your comments raised in a previous round of review and you feel that this manuscript is now acceptable for publication, you may indicate that here to bypass the “Comments to the Author” section, enter your conflict of interest statement in the “Confidential to Editor” section, and submit your "Accept" recommendation.

Reviewer #3: All comments have been addressed

2. Does this manuscript meet PLOS Global Public Health’s publication criteria? Is the manuscript technically sound, and do the data support the conclusions? The manuscript must describe methodologically and ethically rigorous research with conclusions that are appropriately drawn based on the data presented.

Reviewer #3: Yes

3. Has the statistical analysis been performed appropriately and rigorously?

Reviewer #3: Yes

4. Have the authors made all data underlying the findings in their manuscript fully available (please refer to the Data Availability Statement at the start of the manuscript PDF file)?

Reviewer #3: Yes

5. Is the manuscript presented in an intelligible fashion and written in standard English?

Reviewer #3: Yes

6. Review Comments to the Author

Reviewer #3: Dear Authors,

Thank you for the clarification. I accept all your answers. I think the current version of the manuscript merits publication in the journal as an original article.

7. PLOS authors have the option to publish the peer review history of their article (what does this mean?). If published, this will include your full peer review and any attached files.

**Do you want your identity to be public for this peer review?** For information about this choice, including consent withdrawal, please see our Privacy Policy.

Reviewer #3: No
